# Diagnosis of neonatal and adult sepsis using a Serum Amyloid A lateral flow test

Julia Nowak[1], Jacquellyn Nambi Ssanyu[2], Flavia Namiiro[3], Nicola Mountford[4], Avery Parducci[4], Katarina Domijan[5], Mandy Daly[6], Deirdre O'Brien[7], Eithne Barden[7], Kieran Walshe[8], Sean Doyle[1]*, Peter Waiswa[2,9,10]*

**1** Department of Biology, Maynooth University, Maynooth, Ireland, **2** Makerere University School of Public Health and Global Health, Kampala, Uganda, **3** Mulago National Referral Hospital, Kampala, Uganda, **4** School of Business, Maynooth University, Maynooth, Ireland, **5** School of Mathematics and Statistics, Maynooth University, Maynooth, Ireland, **6** Irish Neonatal Health Alliance, Bray, Wicklow, Ireland, **7** Mercy University Hospital, Cork, Ireland, **8** Accuplex Diagnostics Ltd., Maynooth, Ireland, **9** Department of Global Public Health, Karolinska Institute, Solna, Sweden, **10** Busoga Health Forum, Jinja, Uganda

☯ These authors contributed equally to this work.
* pwaiswa@musph.ac.ug (PW); sean.doyle@mu.ie (SD)

**Data Availability Statement:** Complete data underlying the results presented in the study, which contains all collated and anonymized patient information including images of all lateral flow test

## Abstract

Sepsis is the overwhelming immunological response to infection, which if not treated can lead to multi-organ failure, shock and death. Specifically, neonatal sepsis results in 225,000 neonatal deaths globally per annum. Moreover, Uganda experiences one of the highest materno-fetal death rates (62,000 p.a.), with neonatal sepsis deaths at approximately 6,500 p.a.. The difficulty in diagnosing neonatal sepsis lies in the non-specific signs and symptoms associated with sepsis and an absence of definitive sepsis-specific biomarkers. However, serum amyloid A (SAA) detection has potential as a superior biomarker for the diagnosis of probable neonatal sepsis. Herein, in ethically-approved studies we have deployed a competitive lateral flow test (NeoSep-SAA (research-use only)) to detect SAA in whole blood at patient bedside in a resource-limited environment. Results are available within 10 minutes and test format is compatible with small blood volumes available from neonates (5 µl). NeoSep-SAA exhibited a high sensitivity and specificity for diagnosis of adult sepsis, and in neonates showed a sensitivity and specificity of 92% (89%, 95%) and 73% (68%, 77%) with PPV and NPV of 78% (75%, 81%) and 90% (86%, 93%), respectively (n = 714 individuals; 95% CI). NeoSep-SAA showed superior sensitivity for neonatal sepsis over C-Reactive Protein detection (sensitivity: 37%), albeit with some sacrifice of specificity. NeoSep-SAA enabled rapid diagnosis, which combined with minimally-invasive blood withdrawal, was less stressful for neonates. Overall, NeoSep-SAA can readily identify infection/inflammation and has the potential to enable rapid and informed clinical decisions to combat sepsis. This approach has potential to improve neonatal sepsis detection and reduce neonatal mortality in line with United Nations Sustainable Development Goal (SDG) 3.2 objectives.

results, is available from the Corresponding Authors. Alternatively, the data underlying the results presented in this study can be accessed on request sent to the Makerere University School of Public Health Research and Ethics Committee, contact email sphrecadmin@musph.ac.ug.

**Funding:** This work was funded by Science Foundation Ireland (SFI) and Irish Aid (Project codes: 21/FIP/SDG/9917 and 21/FIP/SDG/9917P). Authors initials who obtained funding: SD, PW, NM, FN and MD. The funders played no role in the study design, data collection or analysis, decision to publish or preparation of the manuscript. https://www.sfi.ie/funding/funding-calls/future-innovator-sdg/.

**Competing interests:** The authors have declared that no competing interests exist, except Kieran Walshe who is Chief Technology Officer of Accuplex Diagnostics Limited.

## Introduction

Neonatal sepsis is a major problem resulting in at least 225,000 neonatal deaths globally per annum [1]. United Nations Sustainable Development Goal (UN SDG) 3.2 aims to end preventable deaths of newborns and children under 5 years of age, globally. The challenge to reduce neonatal mortality to 12 per 1,000 live births and under-5 mortality to 25 per 1,000 live births by 2030 may be in part achieved by improved diagnosis of neonatal sepsis in Low-Middle Income Countries (LMIC). In Uganda, over 32,000 neonatal deaths occur per annum (20 per 1000 live births)—one of the highest rates in the world. As sepsis accounts for 20% of this mortality rate, this results in 17 neonatal deaths per day, or approximately 6,500 neonatal deaths per annum [1].

Many births (up to 25%) happen away from centralised care facilities (Uganda Demographic and Health Survey 2016 cited in [2]). Relatively large blood volumes (1–2 ml) are required for existing laboratory tests and patient bedside tests are either unavailable or not used to facilitate detection of neonatal sepsis [1]. In combination, these factors contribute to the unacceptable neonatal mortality rate. Development and deployment of appropriate neonatal-specific sepsis detection systems, in accordance with ASSURED characteristics (Affordable, Sensitive, Specific, User-friendly, Rapid, Equipment-free, Deliverable) [3, 4] would enable screening and support rapid diagnosis and subsequent treatment at the patient bedside, potentially reducing neonatal mortality.

In addition to clinical signs and symptoms, blood culture and a range of biomarkers including C-Reactive Protein (CRP), procalcitonin (PCT), albumin and IL-6 have been proposed and utilised for the diagnosis of neonatal sepsis [5]. However, limitations of the current sepsis 'Gold Standard' test, blood culture, contribute to the difficulties in neonatal sepsis diagnosis. Blood cultures require large volumes of neonatal blood, specialised laboratory facilities and a long waiting period for results. Additionally, blood culture testing for sepsis has an unacceptable low sensitivity and carries a high false negative risk [6]. Serum Amyloid A (SAA) is a well-characterised protein biomarker of sepsis-associated inflammation and infection, and elevated blood SAA levels have been proven to be sensitive and specific for detecting infection in neonates [7–9]. In a study evaluating 15 different biomarkers (ferritin, fibrinogen, granulocyte colony-stimulating factor (G-CSF), interferon (IFN)-γ, IL-1β, -6, -8, -10, macrophage inflammatory protein (MIP)-1β, PCT, resistin, tumor necrosis factor (TNF-α), tissue plasminogen activator-3 and visfatin; n = 15), SAA detection was shown to be superior for the detection of neonatal sepsis [10]. This study included a neonatal cohort comprising 105 individuals (51 healthy controls and 54 neonates with evidence of sepsis (as judged by clinical evidence and/or CRP or IL-6 elevation). It concluded that SAA, of all biomarkers, exhibited the most favourable kinetics regarding the diagnosis of sepsis. Several other studies have demonstrated that SAA provides useful diagnostic support for determination of neonatal sepsis. Krishnaveni et al. [11] observed sensitivity, specificity, positive predictive value and negative predictive values of 95%, 82%, 81% and 95%, respectively. Moreover, there was no significant difference observed for SAA-enabled diagnosis compared to either PCT or high sensitivity CRP (hs-CRP) detection, for culture positive sepsis [12]. Hence the choice of SAA over PCT and CRP as biomarker, with a location-compatible detection system, for our study. Although these emerging studies provide support for SAA utility for neonatal sepsis, all studies conducted to date required instrumentation including multiplex fluorescent immunoassay [10] or latex enhanced immune turbidimetry [12] thereby preventing patient bedside diagnosis and limiting use in resource-limited environments.

While not absolutely specific for sepsis detection just like all other known inflammatory biomarkers, SAA use presents a number of key advantages for use in a LMIC. Firstly, it is a

highly sensitive and dynamic biomarker of infection. Secondly, it is a stable biomarker, detectable by conventional immunological techniques. Thirdly, in normal healthy patients it is either undetectable or present at very low concentrations in blood. SAA can increase approximately 100 to 1000-fold thereby enabling detection by lateral flow test (LFT) technology [13, 14]. Ideally neonatal sepsis detection also requires a patient bedside format that is compatible with small blood volumes available from neonates, and that can be deployed in a resource-limited environment. LFTs which have a requirement for microliter whole blood test volumes, no sample dilution or pre-treatment and operate without need for extra equipment present an ideal solution to these requirements. Moreover, with minimally-invasive blood withdrawal it is least stressful for the neonate, and can also aid rapid identification of infection/inflammation. It has the potential to facilitate a rapid clinical decision in the absence of centralised test facilities or any existing patient-side biochemical analyses, and to initiate antibiotic therapy to combat sepsis. This treatment can reduce neonatal mortality in line with UN SDG 3.2 objectives.

We undertook ethically-approved clinical evaluations involving neonates across Uganda and engaged with stakeholders to determine if LFT technology and SAA detection could aid diagnosis of neonatal sepsis, and prove useful, in resource-limited environments. Our unique collaborative project, which combines technical, qualitative and technology uptake perspectives has focused on the provision, evaluation and implementation of a (i) reliable, (ii) easy to use, (iii) patient-compatible and (iv) patient-side SAA-LFT. This technology has potential to improve the diagnosis of neonatal sepsis in Uganda and beyond, and potentially lead to a reduction in neonatal mortality. Ultimately, better detection of neonatal sepsis through the application of improved diagnosis will aid antibiotic therapy deployment. In the case of negative test results, it may also enable a reduction in the use of antibiotics to help address the issue of antimicrobial resistance.

## Materials and methods

### Lateral flow test device

The NeoSep-SAA lateral flow test, provided by www.accuplexdiagnostics.com, is designed to detect inflammation/infection, two key clinical conditions associated with sepsis. NeoSep-SAA is a competitive semi-quantitative LFT based on the inhibition of gold nanoparticle-labelled IgG [anti-human SAA] interaction with recombinant human SAA printed onto nitrocellulose membranes by SAA in test specimens (Fig 1). A positive result (elevated or high SAA in a human blood sample) prevents line formation on the test membrane, whereas a negative result is observed when normal SAA levels do not prevent gold nanoparticle-labelled IgG [anti-human SAA] interaction with recombinant human SAA and results in line appearance. The NeoSep-SAA test is performed at patient bedside, using 5 μl whole blood obtained by heelprick (neonatal sample) or finger prick (adult patients or children), and takes 10 min to obtain a result. Serum or plasma can also be used but requires 3 μl sample volume. The use of the NeoSep-SAA test on adult human blood for the diagnosis of sepsis was initially assessed with cooperation of Health Innovation Hub Ireland at The Mercy University Hospital (Cork, Ireland).

### Familiarisation study, neonatal sepsis study and clinical specimens

The LMIC evaluations consisted of two phases: firstly, a NeoSep-SAA Familiarisation Study (FS) was undertaken in Uganda, whereby the NeoSep-SAA test was used to test for the presence or absence of SAA involving 450 adult and 50 neonatal blood specimens (5 μl each; n = 500 in total). Secondly, a Neonatal Sepsis Study (NSS) was undertaken involving 955 neonatal blood specimens (5 μl each). Comparative CRP-testing using a laboratory-based immunoturbidimetric assay was carried out where possible. Subjects for Phase I (FS) were enrolled

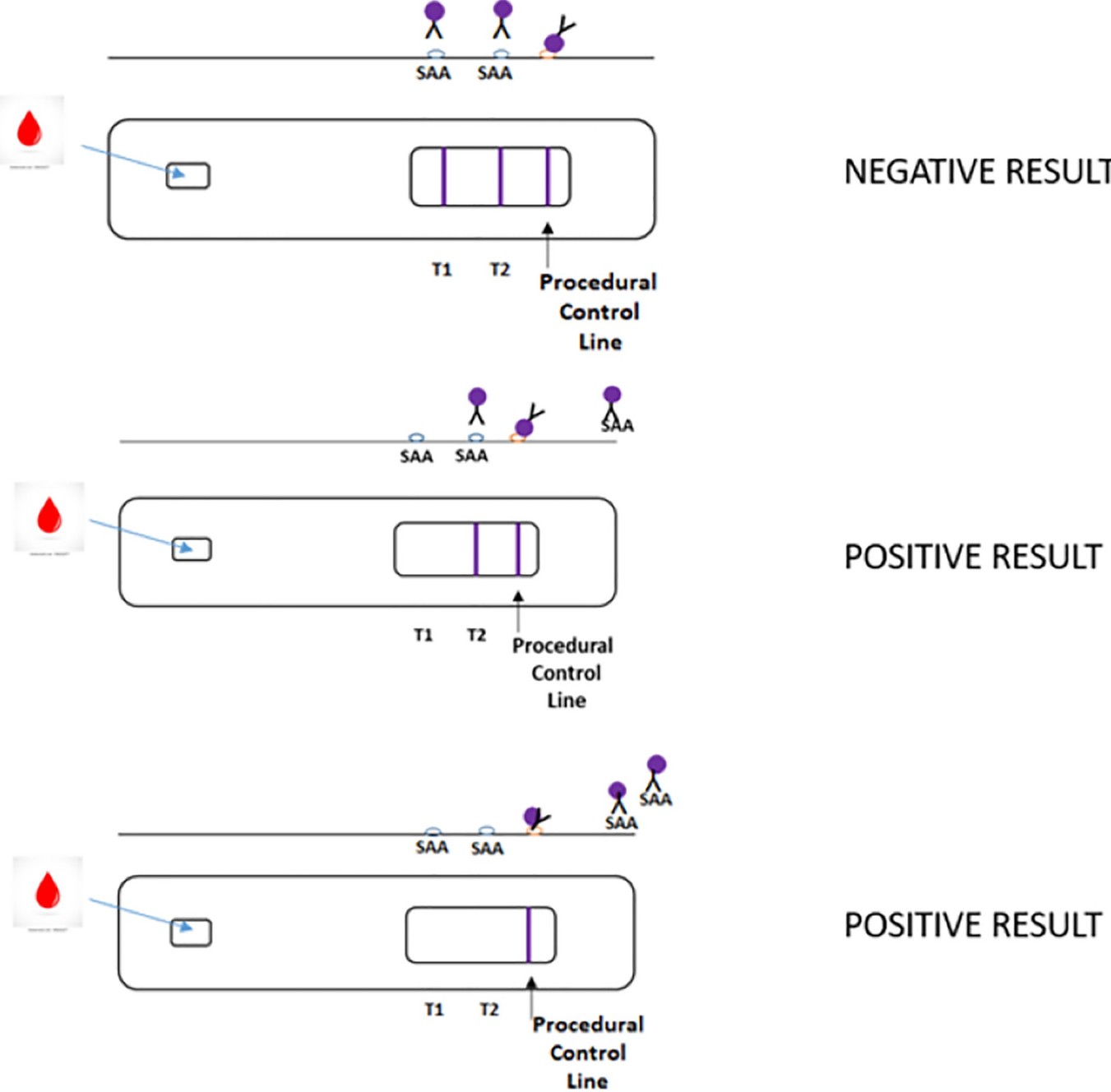

**Fig 1. NeoSep-SAA test schematic depicting a competitive semi-quantitative immunoassay format.** An elevated presence of SAA in human blood samples prevent test line formation by interacting and consequently inhibiting gold nanoparticle-labelled IgG interaction with recombinant human SAA printed on the nitrocellulose membrane. Test lines appear and are interpreted as follows: POSITIVE (1 line) and POSITIVE (2 lines): inflammatory/infection condition, and NEGATIVE (3 lines): No active inflammation.

based on the inclusion criteria: individuals giving blood for diagnostic purposes in selected hospitals; including adult men and women, occasional stored serum or plasma samples for alternative use with permission. To ensure capability of NeoSep-SAA to detect elevated SAA in blood specimens, 66% of blood samples were collected from individuals with inflammation

to yield a positive test result. Neonatal specimens were enrolled in Phase II (NSS) based on the inclusion criteria: clinical specimens obtained from neonatal patients with and without evidence of infection. Patients were excluded from Phase II (NSS) based on the exclusion criteria: neonates on antimicrobial therapy in the last 6 h before presentation to the facility, critically ill infants with a poor outcome prognosis and where a clinical judgement was made that potential participants were too unwell and/or unable to donate a blood specimen.

Overall, 325 hospital-based and 155 field-based clinical neonatal specimens were enrolled in Phase II (NSS) with clinically suspected sepsis (Combined Clinical Group (CCG)). Suspicion of probable sepsis was based on the presentation of $\geq 2$ clinical symptoms of infection, 1 or more clinical symptoms accompanied by a maternal symptom or 1 or more positive laboratory criteria of infection or a neonatal sepsis risk factor. Neonatal clinical and laboratory criteria and risk factors used to indicate probable sepsis are given in Table 1. Sepsis diagnosis was confirmed by Ugandan clinicians based on a suspicion of infection using a combination of clinical symptoms or risk factors aided where possible by a CRP test or microbiological blood cultures.

In parallel, 316 hospital-based and 159 field-based healthy neonatal blood specimens were enrolled in Phase II (NSS) as the Negative Control Group (NCG) if there was no suspicion or evidence of sepsis. Neonates were not suspected to have sepsis if not presenting with sepsis-related clinical symptoms, sepsis-related positive laboratory tests (CRP, CBC) or were admitted to hospital for non-sepsis/infection related reasons. All neonates were subject to full postnatal history taking and clinical evaluation. Any prenatal and natal information was also obtained where available. Baseline parameters for each neonate were recorded (Table 2).

The two studies; FS and NSS, took place in five hospitals providing different levels of care across the Kampala region in Uganda (S1 Table). The field-based study part of NSS was performed in rural/non-hospital locations, defined as a health facility that has no laboratory facilities and where access to health facilities is limited or unavailable.

As part of the FS, training was provided to all staff involved in the use of the NeoSep-SAA test during the study. Training included the correct use of the test and result interpretation which was implemented through written and illustrated instructions supplied with each test kit, a training video in the English, Lusoga and Luganda language, and an online meeting to

**Table 1. Neonatal criteria and neonatal and maternal risk factors used to indicate probable sepsis.**

| Neonatal clinical criteria | Neonatal laboratory criteria | Neonatal and maternal risk factors |
|---|---|---|
| Abnormal temperature (higher than 37.5 ˚C or less than 36.5˚C) or temperature instability | Leukocytosis (white blood cell (WBC) count > 20.0 x $10^9$ cells/L) | Abnormal maternal temperature (> 37.5˚C) |
| Difficulty in feeding or feeding intolerance | Leukopenia (WBC < 4.0 x $10^9$ cells/L) | Prolonged rupture of membranes more than 18 h prior to delivery |
| Abdominal distention | CRP higher than 10 mg/L | Abnormal vaginal discharge for more than 24 h prior to delivery |
| Difficulty in breathing | Abnormal complete blood count (CBC) | Draining liquor more than 24 h prior to delivery |
| Lethargy or drowsiness | | Asphyxia |
| Hypotonia or floppiness | | Suspected early onset neonatal sepsis or infection |
| No movement or movement when stimulated | | Septic rash |
| | | Jaundice |
| | | Seizures |

**Table 2. Baseline parameters of the neonatal cohort from NSS (n = 955).**

| Parameters | (n = 955) |
|---|---|
| **Neonatal** | |
| **Gender**\* | |
| Female | 437 |
| Male | 515 |
| **Gestational Age** | |
| Term: ≥37 weeks | 761 |
| Moderate preterm: 32 to <37 weeks | 130 |
| Very preterm: 28 to <32 weeks | 51 |
| Extremely preterm: <28 weeks | 13 |
| **Mode of Delivery** | |
| Vaginal—spontaneous | 659 |
| Vaginal—Ventouse-assisted | 1 |
| Emergency caesarean section | 276 |
| Elective caesarean section | 19 |
| Mean age (days) at time of testing | 3.8 |
| **Hospital** | |
| Mulago Specialized Women and Neonatal Hospital | 263 |
| Kiwoko | 81 |
| Kawempe National Referral Hospital | 233 |
| Jinja Regional Referral Hospital | 221 |
| Iganga Hospital | 157 |
| **Multiple Birth** | |
| Singleton | 893 |
| Twins | 57 |
| Triplets | 5 |
| **Signs and symptoms of sepsis recorded** | |
| Abdominal distension | 46 |
| Abnormal temperature (higher than 37.5 or less than 36.5) or temperature instability (e.g wide variations) | 242 |
| Difficulty in breathing | 182 |
| Hypotonia/floppiness | 4 |
| Lethargy or drowsiness | 15 |
| Difficulty in feeding or feeding intolerance | 134 |
| No movement or movement when stimulated | 12 |
| **Reason for admission to hospital/ healthcare clinic** | |
| Asphyxia | 103 |
| Jaundice | 68 |
| Respiratory distress | 88 |
| Low birth weight | 15 |
| Prematurity | 147 |
| Feeding difficulties | 112 |
| Seizures | 11 |
| Birth injury | 151 |
| Infection | 32 |
| **Birth facility** | |
| Non-clinical setting | 109 |
| Clinical setting | 846 |

\*Gender was not recorded for 3 neonates.

answer any outstanding questions directly from the staff throughout the five different locations around Uganda (https://www.youtube.com/watch?v=GS1cxGzFaS4). This evaluation was designed to assess the functionality of the NeoSep-SAA to detect SAA in clinical specimens at patient bedside and allowed user training and test familiarisation exercises in Ugandan hospitals which would be involved in a subsequent neonatal sepsis study.

### Statistical analysis

Experimental power calculations were carried out using G*Power statistical package (https://stats.oarc.ucla.edu/other/gpower/), which allows calculation of sample sizes depending on a range of parameters. Specifically, t-test computations to establish the difference between two independent means (two groups) were carried out using the following variables: (i) One-tailed t-test was chosen for the present study as it was known that SAA levels would be higher in clinical groups compared to controls; (ii) Expected effect size d between control and test groups (0.8, large; 0.5, medium and 0.2, small) and (iii) Power values 0.8 and 0.95 (S2 Table). Confidence limits were calculated for test sensitivity, specificity, positive and negative predictive values according to [15].

### Ethical approval

Overall project ethical permission was secured at Makerere University School of Public Health (Reference: SPH-2022-220; duration 17 June 2022–17 June 2023) and Maynooth University (Reference: BSRESC-2022-2477046; duration 24 June 2022–30 June 2023), respectively. The study was also registered with the Uganda National Council of Science and Technology, registration number HS2439ES (Approval date 3 October 2022; duration 3 October 2022–3 October 2024) (S1 Table).

## Results

Prior to FS and NSS, preliminary analysis demonstrated that the NeoSep-SAA could (i) distinguish normal (n = 30; S3 Table) versus elevated SAA levels in adult human clinical blood specimens and (ii) correctly identify elevated SAA (n = 20) in clinical blood specimens obtained from individuals with confirmed sepsis. These data (S3 Table) confirm Neo-Sep-SAA use in an adult human population and its capacity to identify normal versus elevated blood SAA levels, in an Irish population (n = 20 human clinical specimens). Moreover, the NeoSep-SAA correctly identified absence of infection and inflammation (100% concordance) following comparative analysis of normal human sera (n = 30) in parallel analysis against an SAA-ELISA, previously calibrated using the World Health Organisation (WHO) SAA International Standard (IS) (Code 92/680) (S3 Table).

### Familiarisation study

The FS confirmed the functionality of the NeoSep-SAA to detect SAA in clinical specimens (n = 500) at patient bedside (S4 Table). The FS allowed user training and test familiarisation exercises in Ugandan hospitals which were involved in the subsequent NSS. Once it was clear that the NeoSep-SAA was functioning as expected, preliminary testing commenced using 50 neonates in the FS. Minimal positive results were obtained for healthy patients in the FS (S4 Table). One other key message that derived from the FS centred around the definition of 'healthy' patients and resulted in the unambiguous and agreed definition of healthy patients for the NSS as those with no known underlying clinical or infectious disease conditions in negative control patients.

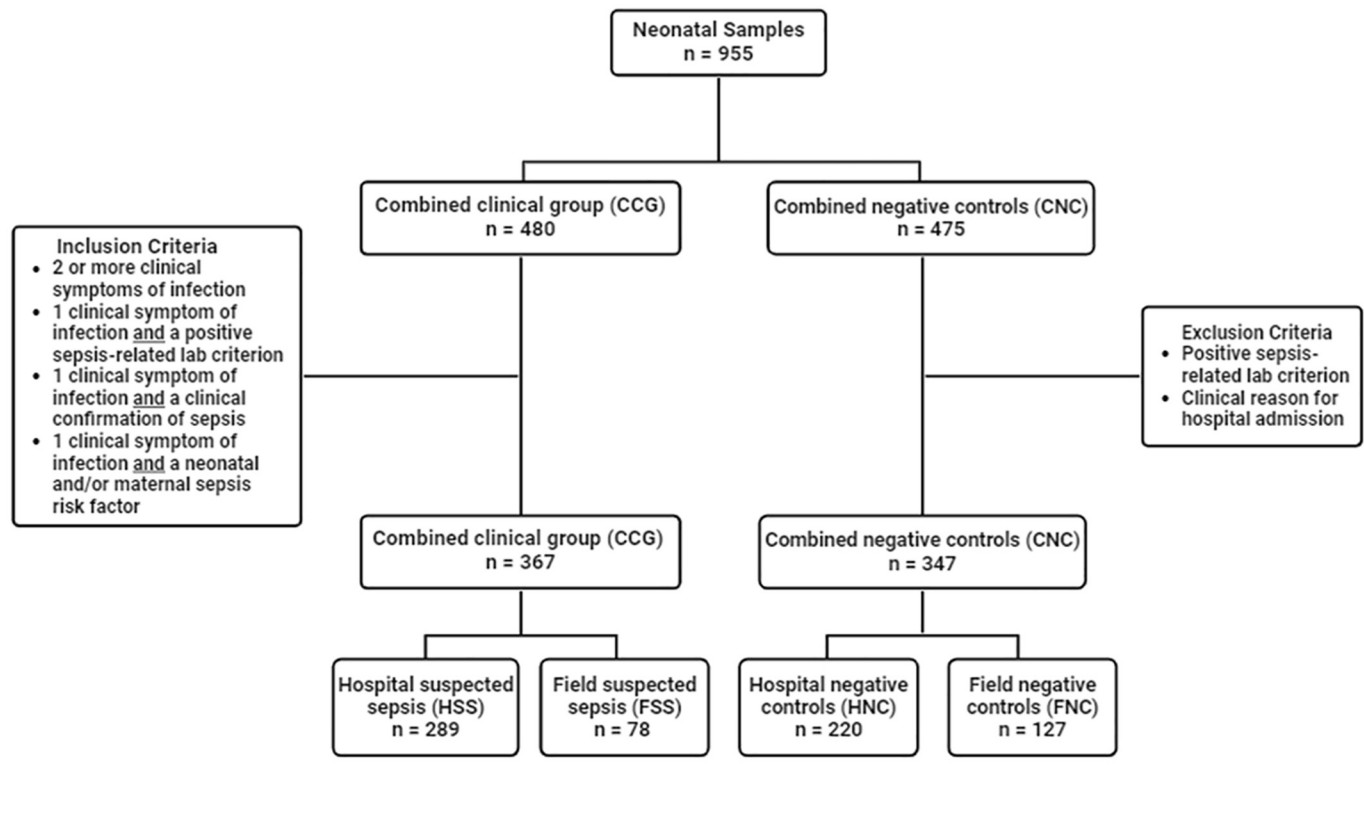

**Fig 2. Flowchart depicting neonatal sample dataset categories as well as the application of inclusion and exclusion criteria on the dataset.**

### Neonatal sepsis study

As per the inclusion and exclusion criteria outlined above, the final dataset of the NSS, consisted of n = 714/955 neonatal samples across the hospital- and field-based clinical and control populations (Fig 2). The NSS involved the patient bedside testing of blood specimens by Neo-Sep-SAA from (i) 220 hospital negative control (HNC) neonates, (ii) 289 hospital-based neonates with evidence or diagnosis of sepsis (hospital suspected sepsis: HSS), (iii) Blood specimens were also collected for testing by NeoSep-SAA from neonates outside of a hospital setting, comprising 127 neonates with no evidence of sepsis (field negative controls: FNC) and 78 field-based neonates with evidence or diagnosis of sepsis (field suspected sepsis: FSS). Hereafter HSS and FSS in combination are referred to as Combined Clinical Group (CCG), similarly, HNC and FNC in combination are referred to as Combined Negative Controls (CNC).

Overall, males comprised 379/714 (53%) of the total neonatal cohort for the NSS. No gender information was available for 3 neonates. Males comprised 146/289 (50%) of hospital-based suspected sepsis group (gender unknown 1%). Males comprised 112/220 (51%) of the negative control neonate population (no gender information available for 1 neonate). Thus, both groups were gender-balanced and NeoSep-SAA performed equivalently in males and females.

Overall, 338/367 (92%) neonates in the CCG tested positive in the NeoSep-SAA and 252/347 (73%) neonates in the CNC tested negative in the NeoSep-SAA. This translates to an

**Table 3. NeoSep-SAA and comparator CRP test performance data from neonatal sepsis study (CCG vs. CNC, n = 714) in terms of sensitivity, specificity, PPV and NPV.**

|  | Sensitivity % | Specificity % | PPV % | NPV % | Accuracy % |
|---|---|---|---|---|---|
| **NeoSep-SAA** | 92 | 73 | 78 | 90 | 83 |
| **CRP test** | 37 | 100 | 100 | 54 | 64 |

overall sensitivity and specificity of the NeoSep-SAA for screening or probable diagnosis of neonatal sepsis of 92% (89%, 95%) and 73% (68%, 77%), at 95% CI, respectively. Resultant positive predictive value (PPV) and negative predictive values (NPV) equate to 0.78 (0.75, 0.81) and 0.90 (0.86, 0.93), respectively (Table 3). Only 107/367 (29%) neonates in CCG tested positive in the comparator CRP test, resulting in sensitivity and specificity of 37% (31%, 42%) and 100% (98%, 100%), at 95% CI, respectively (Table 3). High specificity of the CRP test is attributable to the use of a positive CRP test as an excluding factor in the control populations.

Additionally, 78 neonates within the CCG were clinically diagnosed with sepsis (Confirmed Sepsis: CS). Sepsis diagnosis was based on a suspicion of infection and combination of clinical symptoms or risk factors aided where possible by a CRP test, blood cultures and confirmed by an independent clinician's diagnosis. Thus, these 78 neonates were clinically diagnosed with sepsis by clinicians independent of the NeoSep-SAA result. Of the 78 neonates, 72 (92%) neonates with a confirmed sepsis diagnosis tested positive in the NeoSep-SAA (Table 4).

It was important to assess the performance of the NeoSep-SAA test in non-clinical settings to ensure the test can also be used bedside in rural environments where laboratory facilities are most limited. Field-based data was therefore separately assessed and the performance of the NeoSep-SAA in the solely field-based cohort translated to a sensitivity and specificity of 92% (84%, 97%) and 72% (64%, 80%), at 95% CI, respectively (Table 4). It is important to note that CRP testing was not performed in the field-based NSS cohort due to its unavailability. The test performance was also evaluated for the full NSS neonatal cohort, n = 955, to assess the test independently of the criteria (Table 5). Albeit a reduced specificity at 63% (58%, 67%), the performance remained largely unchanged with a sensitivity of 91% (88%, 94%), PPV of 71% (69%, 74%) and NPV of 88% (84%, 91%), at 95% CI (Table 5).

## Discussion

Neonatal sepsis disproportionately burdens LMICs, is responsible for 4 of every 20 neonatal deaths and failure to diagnose is a significant contributory factor. This absence of robust, sensitive and location-appropriate diagnostic systems creates an urgent need for evidence-based decision-making facilitated by rapid, onsite clinical test results. LFT technology is attracting ever-increasing attention for the detection of altered clinical conditions or host infection status [16, 17]. Herein, work focused on assessing NeoSep-SAA, a novel LFT device for detection of SAA, to help improve both diagnosis of probable sepsis in neonates and hopefully contribute

**Table 4. NeoSep-SAA performance data from neonatal sepsis study sub-groups, in terms of sensitivity, specificity, PPV and NPV.**

| Population | Sensitivity % | Specificity % | PPV % | NPV % | Accuracy % |
|---|---|---|---|---|---|
| **HSS vs. HNC** | 92 | 73 | 82 | 87 | 84 |
| **FSS vs. FNC** | 92 | 72 | 67 | 94 | 80 |
| **CS vs. CNC** | 92 | 73 | 43 | 98 | 76 |

HSS: Hospital Suspected Sepsis; HNC: Hospital Negative Controls; FSS: Field Suspected Sepsis; FNC: Field Negative Controls, CS: Confirmed Sepsis; CNC: Combined Negative Controls.

**Table 5. NeoSep-SAA primary performance data from NSS (n = 955).** Comparative NeoSep-SAA and CRP test performance in terms of Sensitivity, Specificity, PPV and NPV in full NSS combined field and hospital cohort.

| | Sensitivity % | Specificity % | PPV % | NPV % | Accuracy % |
|---|---|---|---|---|---|
| **NeoSep-SAA** | 91 | 63 | 71 | 88 | 77 |
| **CRP test** | 32 | 91 | 78 | 58 | 62 |

to a reduction in neonatal mortality, a key objective of UN SDG 3.2. Specifically, we describe the development and evaluation of a competitive LFT for the detection of human SAA, as a screening test with potential to predict sepsis, specifically neonatal sepsis, in resource limited environments. As part of a preliminary study we demonstrated that NeoSep-SAA can detect SAA and predict sepsis in an adult population when compared to a quantitative SAA ELISA (calibrated against the WHO SAA IS), CRP and microbiological culture. In a subsequent highly powered study comprising a large neonatal cohort (n = 714), we demonstrate NeoSep-SAA performance and suitability in a LMIC for detection of probable neonatal sepsis with high sensitivity and specificity, 92% and 73%, respectively. Given both the unsuitability of current sampling systems and absence of current testing for neonatal sepsis in LMICs, we propose that NeoSep-SAA is ideal for supporting the detection of probable neonatal sepsis in LMICs.

Detection and treatment of sepsis in newborn babies and infants is difficult given that symptoms are non-specific and can be readily confused with other clinical conditions. In LMICs detection of sepsis is complicated by a lack of available diagnostic tests and uncertain criteria used for making clinical decisions [18]. In the absence of a diagnostic test and lack of available laboratory facilities in certain hospital and non-clinical settings, this is even more difficult. The use of multiple clinical symptoms and/or maternal and neonatal risk factors represents the reality of assessing neonates for suspicion of sepsis in resource-limited environments. The issue is further complicated by the many differing definitions of sepsis, and the fact that those that apply to the adult population do not necessarily apply to the neonate [19].

An early-stage indication of probable sepsis is based on suspicion of infection combined with clinical symptoms and/or neonatal risk factors such as premature birth or maternal risk factors [20]. As many samples during this study were taken in remote locations with no access to laboratory testing, identification of neonates with suspected sepsis was based solely on clinical symptoms, as seen previously [21]. New definitions of sepsis tend to disregard symptoms of systemic inflammatory response syndrome (SIRS) as indicators of a problem. However, since sepsis involves a dysregulated inflammatory response, inflammatory biomarkers can provide a clear indication of probable sepsis, alongside other existing symptoms. Furthermore, in the neonate, several studies have shown that there is a high correlation between SIRS and sepsis, with a particularly high sensitivity [22, 23]. In addition, multiple studies have demonstrated that SAA, a key indicator of a systemic inflammatory condition associated with infection, has both a high sensitivity, and in well controlled studies based in a hospital setting, a high specificity in facilitating diagnosis of neonatal sepsis [12, 24, 25].

Prior to deployment of the NeoSep-SAA test for SAA testing to detect neonatal sepsis, it was subjected to two preliminary evaluations, one in an Irish population (n = 50; 30 negative controls; 20 clinical specimens) and one in a Ugandan population consisting of 500 individuals. The Irish study confirmed test functionality for SAA detection and discrimination between a control and sepsis-diagnosed cohort with confirmed infection. The Ugandan FS further supported test performance and facilitated training of clinical and midwifery staff in test performance, interpretation and data recording. In the neonatal CCG group 338/367 (92%) patients tested positive, while 252/347 (73%) neonates in the CNC tested negative in the NeoSep-SAA.

The resultant sensitivity and specificity of the NeoSep-SAA for diagnosis of probable neonatal sepsis was 92% (89%, 95%) and 73% (68%, 77%), at 95% CI, respectively, with PPV and NPV of 0.78 (0.75, 0.81) and 0.90 (0.86, 0.93), respectively (Table 3).

In contrast to SAA detection, CRP, an established biomarker of infection, presented with a surprisingly low sensitivity of only 37% (31%, 42%) at 95% CI. Positive CRP measurements are the current, extensively used tests to facilitate confirming the suspicion of sepsis, despite the concerning low sensitivity as seen not only in this study and others [26, 27]. The risk of over-looking a high number of septic neonates and reducing their chance of diagnosis, treatment, and prognosis should be unacceptable. In this study, NeoSep-SAA significantly out-performed the currently available gold-standard biomarker, CRP, for neonatal sepsis detection (p<0.0001, 95% CI). It has been shown that serum lactate >4 mmol/L (OR = 3.4), amongst other clinical criteria, was significantly (p<0.05) associated with mortality due to neonatal sepsis [28]. Moreover, blood lactate levels have recently attracted additional attention as a putative biomarker of early-onset clinical sepsis in neonates, with > 3.38 mM venous blood lactate, present 6 h post-natally, proposed as a cut-off value [29]. However, while lactate is a useful biomarker for supporting diagnosis of sepsis, it may lack sensitivity and future work involving both lactate and SAA determination may lead to further improvements in neonatal sepsis detection.

A comprehensible test such as the NeoSep-SAA which can be performed by general staff with basic training is highly appropriate for integration into lower-level health facilities which deliver the highest volumes of neonatal care. Feedback from healthcare staff in Uganda during this study revealed that many deliveries occur in lower-level health care facilities, such as Health Care Centre III or IV, or indeed outside the healthcare system. In such contexts, a lack of required equipment, specialized staff, and necessary resources prevents use of routine tests such as blood cultures or the CRP test for sepsis identification. Furthermore, although higher-level facilities may have the means for routine laboratory-based tests; restrictions in terms of inadequately trained staff, understaffing, overcrowding and economic barriers limits the use of these tests. One significant downside to this is the widespread prescription of antibiotics to neonates within the hospital. The feedback further affirmed the ease-of-use and satisfaction with the low-invasive nature, rapidness and useability of the NeoSep-SAA test. This feedback further supports the suggested adoption of the NeoSep-SAA into routine neonatal testing in aid of sepsis detection with potential to reduce the unnecessary use of antibiotics or if SAA detection is negative, terminate antibiotic therapy. Given that Uganda is currently accelerating investment to reach UN SDG targets for reducing neonatal sepsis mortality and has significant experience in operating LFT technology, the availability of NeoSep-SAA is timely [30]. Moreover, the NeoSep-SAA test could readily be adapted as part of Integrated Management of Newborn and Childhood Illnesses (IMNCI) [31].

A significant advantage of using the NeoSep-SAA is the requirement of a very small heel prick blood sample. A minimally invasive sampling procedure is not only less painful and less stressful but also carries a lower risk of infection to the neonate and avoids the need for use of aseptic technique, associated with more invasive procedures. Interestingly, given the serious issue of antimicrobial resistance observed in LMIC, a high NPV would be very helpful in antibiotic stewardship [32, 33]. Neo-Sep SAA had an NPV as high as 98% (Table 4). Therefore, in the absence of clear clinical symptoms, a negative test would support the decision to withhold antibiotics while continuing observation of the neonate. The nature of the NeoSep-SAA would allow the option of a retest within 12–24 h to reassess the condition of the neonate where there may be some suspicion of infection pending.

Future work will assess the utility of the NeoSep-SAA to monitor antibiotic therapy in neonatal sepsis. This planned study will incorporate comparator CRP testing and blood cultures

to provide conclusive results on NeoSep-SAA performance. If successful, it would aid clinicians to make decisions regarding continuation, adjustment or termination of antibiotic treatment. Reducing the excessive use of antibiotics would lower hospitalisation expenditures, decrease the occurrence of stock outs, and prevent the development of multi-drug-resistant microbes.

## Conclusion

It is widely recognised that a definitive test for sepsis does not yet exist, due to the complexity of the condition, lack of sepsis-specific biomarkers and variable symptoms [23]. Additionally, existing tests for sepsis are impractical, especially in resource-limited environments, and lack diagnostic accuracy. However, the introduction of a functional, bedside, rapid test with high sensitivity, good diagnostic accuracy and sufficient specificity would provide critical assistance for the diagnosis of probable neonatal sepsis. This is particularly important in environments where the lack of resources hinders neonatal care. This study has demonstrated that the performance of NeoSep-SAA has proven more satisfactory than the comparator, CRP test, based on statistical analysis, and by contextual appropriateness.

## Supporting information

**S1 Table. Hospitals involved in the familiarisation and neonatal sepsis studies.** Subsequently to acquisition of ethical permission, local administrative clearance was secured from each of the 5 participating Ugandan hospitals. Clinical specimens were collected from patients and data was accessed for research purposes between 7 November 2022 and 18 May 2023. All adult participants in FS provided written informed consent. For NSS, parents or guardians provided written informed consent for neonates. For participants who were unable to read or write, consent was obtained in the presence of a witness who read and explained the ethics statement. Participants confirmed their consent by affixing their thumbprints to the consent forms, which were also countersigned by the witness. Authors did not, and do not, have access to information that could identify individual participants after data collection.
(PDF)

**S2 Table. Calculation of Sample Size N and corresponding test and control and test group numbers\*.** Significance level $\alpha$ = 0.05 was set in all cases and an equal number of patients in each comparison group was required. Resultant Power calculations using G\*Power, based on a small result difference (0.2) and power factor 0.8, yielded a study population size n = 620 (n = 310 normals and n = 310 sepsis patients) for the hospital-based study and a population size n = 620 (n = 310 normals and n = 310 sepsis patients) for the field-based study. However, it should be noted that these participant requirement numbers also exceeded those required for medium (0.5) and large (0.8) expected differences at power factor 0.95. This is based on the possibility that a spectrum of SAA levels will be detected in the sepsis groups, compared to low levels in the control groups. Thus, all resultant experimental data was highly powered.
(PDF)

**S3 Table. SAA lateral flow test results.** SAA lateral flow test results[a] for normal human sera (n = 30) and infected human sera (n = 20) compared to parallel SAA-ELISA[b,c] quantification of SAA levels.
(PDF)

**S4 Table. NeoSep SAA results from the familiarisation study.** Clinical specimens.
(PDF)

## Acknowledgments

Special thanks to all Stakeholders involved across Ireland and Uganda, as well as everyone who participated in sample and data collection including all patients, families, nurses, clinicians and laboratory scientists. The Health Innovation Hub Ireland (HIHI) is acknowledged and thanked for assisting with assessment of NeoSep-SAA at the Mercy University Hospital.

## Author Contributions

**Conceptualization:** Flavia Namiiro, Nicola Mountford, Mandy Daly, Kieran Walshe, Sean Doyle, Peter Waiswa.

**Data curation:** Julia Nowak, Jacquellyn Nambi Ssanyu.

**Formal analysis:** Julia Nowak, Jacquellyn Nambi Ssanyu, Flavia Namiiro, Katarina Domijan, Deirdre O'Brien, Sean Doyle, Peter Waiswa.

**Funding acquisition:** Flavia Namiiro, Nicola Mountford, Mandy Daly, Sean Doyle, Peter Waiswa.

**Investigation:** Flavia Namiiro, Avery Parducci, Mandy Daly, Sean Doyle, Peter Waiswa.

**Methodology:** Julia Nowak, Jacquellyn Nambi Ssanyu, Flavia Namiiro, Nicola Mountford, Avery Parducci, Katarina Domijan, Deirdre O'Brien, Eithne Barden, Kieran Walshe, Sean Doyle, Peter Waiswa.

**Project administration:** Julia Nowak, Peter Waiswa.

**Resources:** Katarina Domijan, Deirdre O'Brien, Eithne Barden, Kieran Walshe.

**Supervision:** Flavia Namiiro, Sean Doyle, Peter Waiswa.

**Validation:** Jacquellyn Nambi Ssanyu, Katarina Domijan, Sean Doyle.

**Writing – original draft:** Julia Nowak, Jacquellyn Nambi Ssanyu, Flavia Namiiro, Kieran Walshe, Sean Doyle, Peter Waiswa.

**Writing – review & editing:** Julia Nowak, Jacquellyn Nambi Ssanyu, Flavia Namiiro, Nicola Mountford, Avery Parducci, Katarina Domijan, Mandy Daly, Deirdre O'Brien, Eithne Barden, Kieran Walshe, Sean Doyle, Peter Waiswa.

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
