## [Decision Letter · Decision Letter 0]

23 Aug 2024

PONE-D-24-29224Diagnosis of neonatal and adult sepsis using a Serum Amyloid A lateral flow test.PLOS ONE

Dear Dr.
Doyle,

Thank you for submitting your manuscript to PLOS ONE. After careful consideration, we feel that it has merit but does not fully meet PLOS ONE’s publication criteria as it currently stands. Therefore, we invite you to submit a revised version of the manuscript that addresses the points raised during the review process.

**ACADEMIC EDITOR:**

**Please address all issues raised by the reviewers. Pay attention to the comments related to the methods section and ensure that the conclusions are an accurate reflection of any new changes made. Please carefully proof-read your manuscript before submitting the revised version.**

We look forward to receiving your revised manuscript.

Kind regards,

Novel N. Chegou, Ph.D

Academic Editor

PLOS ONE

Journal requirements: 1. When submitting your revision, we need you to address these additional requirements. Please ensure that your manuscript meets PLOS ONE's style requirements, including those for file naming. The PLOS ONE style templates can be found at https://journals.plos.org/plosone/s/file?id=wjVg/PLOSOne_formatting_sample_main_body.pdf and https://journals.plos.org/plosone/s/file?id=ba62/PLOSOne_formatting_sample_title_authors_affiliations.pdf. 2. Thank you for stating the following in the Acknowledgments Section of your manuscript: [This work was funded by Science Foundation Ireland (SFI) and Irish Aid (Project codes: 21/FIP/SDG/9917 and 21/FIP/SDG/9917P). Special thanks to all Stakeholders involved across Ireland and Uganda, as well as everyone who participated in sample and data collection including all patients, families, nurses, clinicians and laboratory scientists. The Health Innovation Hub Ireland (HIHI) is acknowledged and thanked for assisting with assessment of NeoSep-SAA at the Mercy University Hospital.]We note that you have provided funding information that is not currently declared in your Funding Statement. However, funding information should not appear in the Acknowledgments section or other areas of your manuscript. We will only publish funding information present in the Funding Statement section of the online submission form. Please remove any funding-related text from the manuscript and let us know how you would like to update your Funding Statement. Currently, your Funding Statement reads as follows:  [This work was funded by Science Foundation Ireland (SFI) and Irish Aid (Project codes: 21/FIP/SDG/9917 and 21/FIP/SDG/9917P). Authors initials who obtained funding: SD, PW, NM, FN and MD.The funders played no role in the study design, data collection or analysis, decision to publish or preparation of the manuscript. https://www.sfi.ie/funding/funding-calls/future-innovator-sdg/] Please include your amended statements within your cover letter; we will change the online submission form on your behalf. 3. In the online submission form, you indicated that [The data underlying the results presented in the study are available from the Corresponding Authors.]. All PLOS journals now require all data underlying the findings described in their manuscript to be freely available to other researchers, either 1. In a public repository, 2. Within the manuscript itself, or 3. Uploaded as supplementary information.This policy applies to all data except where public deposition would breach compliance with the protocol approved by your research ethics board. If your data cannot be made publicly available for ethical or legal reasons (e.g., public availability would compromise patient privacy), please explain your reasons on resubmission and your exemption request will be escalated for approval.  4. Please include captions for your Supporting Information files at the end of your manuscript, and update any in-text citations to match accordingly. Please see our Supporting Information guidelines for more information: http://journals.plos.org/plosone/s/supporting-information. 

Reviewers' comments:

Reviewer's Responses to Questions

**Comments to the Author**

1. Is the manuscript technically sound, and do the data support the conclusions?

Reviewer #1: Yes

Reviewer #2: Partly

2. Has the statistical analysis been performed appropriately and rigorously? 

Reviewer #1: Yes

Reviewer #2: I Don't Know

3. Have the authors made all data underlying the findings in their manuscript fully available?

Reviewer #1: Yes

Reviewer #2: Yes

4. Is the manuscript presented in an intelligible fashion and written in standard English?

Reviewer #1: Yes

Reviewer #2: No

5. Review Comments to the Author

Reviewer #1: Reviewer comments

Manuscript title: Diagnosis of neonatal and adult sepsis using a Serum Amyloid A lateral flow test.

Manuscript number: PONE-D-24-29224

Abstract

1. Page 2, line 35: Author states (AS)…’The difficulty in diagnosing neonatal sepsis……. sepsis-specific biomarkers.’

Reviewer comment (RC): The greatest difficulty is that the ‘Gold Standard’ for diagnosis of sepsis i.e. Blood culture is not always positive even in the presence of true sepsis and the long time it takes for the Blood culture result to be available even with BACTEC/BacTAlert.

A mention of this limitation must be made in the Introduction.

2. Page 2, line 38: AS….’as a superior biomarker for the diagnosis of neonatal sepsis.’

RC: Acute phase reactants can only be used to indicate the probable diagnosis of sepsis or the absence of of sepsis (High NPV) in a neonate with commensurate clinical symptoms and signs. They are not a definitive test for diagnosis of sepsis. Throughout the manuscript, authors have mentioned the role of SAA as a diagnostic test. Even though I do not want to take away the enthusiasm of the research team and the authors, the manuscript must be modified where it is made abundantly clear that SAA is only a novel biomarker for the probable diagnosis of neonatal sepsis.

3. Page 2, line 40: AS…. blood at patient-side….

RC: … at patient bedside….

4. Page 2, line 40: AS….low resource environment…

RC: I suggest that ‘resource constrained’ or ‘resource limited’ be used throughout the manuscript rather than ‘low resource’.

5. Page 2, line 46: AS…..(sensitivity: 37 %)…’.

RC: This sensitivity for CRP was as noted in the study cohort. However, abundant published literature has noted the sensitivity to be in the range as reported for the SAA.

6. Page 2, line 51: AS….SDG 3.2 objectives.

RC: SDG can be expanded for the first time and subsequently can be mentioned as SDG throughout the text.

Introduction

1. The introduction is rambling and requires to be made more focussed.

2. Why did the authors select SAA over Serum Procalcitonin and CRP?

3. Colorimetric card tests are available for CRP, PCT and are very efficiently being used in field out of hospital settings. Then why SAA?

4. In a suspected sepsis patient one would be collecting samples for Complete Blood Counts (CBC) and Blood culture. Thus minimally invasive sampling for SAA wold not be much of a benefit.

5. In most neonatal units in neonates suspected to be having sepsis based on clinical features, antibiotic therapy would be initiated. Are the authors implying that use of SAA can limit starting antibiotics in symptomatic neonates before the results of blood/CSF/ urine culture are available. I think not. Thus the introduction must be about the ability of SAA to help a clinician stop antibiotics early.

6. The hypothesis is not clearly stated.

7. The aim of the study must be stated in the last paragraph of the introduction.

Material & methods

1. Page 6, line 143-145: AS….’up to 66% of patient specimens and donors should have evidence of inflammation to indicate presence of SAA in said test specimens’.

RC: How was this figure of 66% positive patient samples decided?

2. Page 7, line 153: AS…’Suspicion of sepsis….’

RC: See comment Sr no. 2 for Abstract section.

Page 7, line158: AS….’factors represents the reality of assessing neonates…’

RC: This statement is suited in the introduction/discussion section and not in the methods section.

3. Page 6, line 169: AS…..suspected infection…..’.

RC: This has been stated in the sentence and seems like a duplication.

Results

1. Page 12, line 276-277: AS….’sensitivity and specificity of the NeoSep-SAA for diagnosis of neonatal sepsis of 92% (89%, 95%) and 73% (68%, 77%), at 95% CI, respectively’.

RC: The high sensitivity and NPV and lower specificity and PPV indicate towards a role as a screening test for neonatal sepsis.

2. Page 12, line 283: AS….’ Additionally, 78 neonates within the CCG were clinically diagnosed with sepsis (Confirmed Sepsis: CS).’

RC: Confirmed sepsis can only be neonates where Blood/urine/CSF culture yielded growth of a bacteria not likely to be a contaminant. For all other cases the diagnosis can be probable sepsis.

Discussion and conclusion

1. These will have to be modified based on the comments given above.

Reviewer #2: Major

Overall

1. Proofread for grammatical errors, typos, and inconsistencies. Ensure that the language is formal and appropriate for a scientific audience.

Introduction

1. Several sentences are long and complex, making them difficult to follow. Breaking them into shorter sentences can improve readability and clarity.

2. The introduction uses both "per annum" and "annually." Consistency in terminology is essential. Use one term consistently.

3. Ensure that technical terms and jargon are explained clearly for readers who may not be familiar with them.

4. Clearly state the objectives and aims of the study in relation to the problem outlined. This helps in setting up the context for the research.

Methodology:

1. Its too long

2. Define terms and abbreviations when first introduced and ensure they are used consistently. For instance, “SAA” should be defined as “Serum Amyloid A” when first mentioned.

3. Break down long or complex sentences into shorter, clearer ones. This will improve readability and comprehension.

4. The text often repeats phrases or concepts, such as "SAA lateral flow test," which can make the content seem redundant. Recommendation is to streamline repetitive language and consolidate similar information to enhance readability and maintain focus.

5. The text sometimes shifts terminology without clear definition, such as switching between "hospital negative control" and "combined negative controls.Define and consistently use terminology throughout the document. Ensure that all abbreviations and terms are clearly defined upon their first use.

6. Some sentences are overly complex or informal, which can hinder understanding. Phrases like "gives the green light" are less formal and can be replaced with more precise language. Rephrase sentences to be more formal and concise. Break up long sentences into shorter, clearer statements.

Discussion:

1. The discussion is lengthy and contains complex sentences, which may overwhelm readers. Breakdown lengthy sentences into shorter, more digestible parts. Ensure each paragraph conveys a single main idea or argument.

2. Some points are repeated, such as the performance metrics of NeoSep-SAA compared to CRP. Avoid repetition by consolidating similar points and emphasizing unique findings in different sections.

3. The use of technical terms and abbreviations may be difficult for non-specialist readers to understand. Define technical terms and abbreviations upon first use and consider adding a glossary or simplifying explanations where possible.

4. While the discussion includes relevant information, some background details on why the test is needed are not fully fleshed out. Provide more context about the global burden of neonatal sepsis and the limitations of current diagnostic methods to highlight the importance of NeoSep-SAA.

5. While the potential impact of NeoSep-SAA is mentioned, the discussion could better explore its practical applications and limitations.

6. The section on future work is somewhat brief and lacks specific suggestions for further research. Elaborate on potential future studies, including how NeoSep-SAA might be combined with other diagnostic tools or tested in different populations.

Minor

1. Line 55 – spell out UN SDG

2. Line 60 - https://www.unicef.org/uganda/research-and-reports add the reference as well.

3. Line 61-65 – sentence is too long

4. Line 65 – these contribute….. consider these factors contribute

5. Line 67 – ASSURED should be (Affordable, Sensitive, Specific, User-friendly, Rapid and robust, Equipment-free and Deliverable)

6. Line 68 – reference should be like [3,4]

7. Line – 118 - NeoSep-SAA – spell out

8. Line 118 - the URL inclusion is not standard in the body of the text. URLs should be included in the references section or integrated into the text more seamlessly.

9. Line 147 – informed consent……should be in ethics sub heading

10. Line 243 – WHO reference

11. Line 250-252 - sentence is lengthy and could benefit from breaking into two for better readability.

12. Line 299 – not good to start the sentence with an abbreviation

13. Line 316-318 – rephrase if for clarity

6. PLOS authors have the option to publish the peer review history of their article (what does this mean?). If published, this will include your full peer review and any attached files.

Reviewer #1: **Yes: **Vishal Vishnu Tewari

Reviewer #2: No

---

## [Author Response · Author response to Decision Letter 0]

25 Oct 2024

Manuscript title: Diagnosis of neonatal and adult sepsis using a Serum Amyloid A lateral flow test.

Manuscript number: PONE-D-24-29224

RESPONSE TO REVIEWERS

ACADEMIC EDITOR:

Please address all issues raised by the reviewers. Pay attention to the comments related to the methods section and ensure that the conclusions are an accurate reflection of any new changes made. Please carefully proof-read your manuscript before submitting the revised version.

 Author response: Noted and done. All issues raised by Reviewers have been comprehensively addressed, especially Methods. Conclusions accurately reflect changes made. Manuscript R1 has been extensively proof-read prior to re-submission.

JOURNAL REQUIREMENTS:

 Author response: Noted and done.

[This work was funded by Science Foundation Ireland (SFI) and Irish Aid (Project codes: 21/FIP/SDG/9917 and 21/FIP/SDG/9917P). Special thanks to all Stakeholders involved across Ireland and Uganda, as well as everyone who participated in sample and data collection including all patients, families, nurses, clinicians and laboratory scientists. The Health Innovation Hub Ireland (HIHI) is acknowledged and thanked for assisting with assessment of NeoSep-SAA at the Mercy University Hospital.]

 [This work was funded by Science Foundation Ireland (SFI) and Irish Aid (Project codes: 21/FIP/SDG/9917 and 21/FIP/SDG/9917P). 

Authors initials who obtained funding: SD, PW, NM, FN and MD.

The funders played no role in the study design, data collection or analysis, decision to publish or preparation of the manuscript. 

https://www.sfi.ie/funding/funding-calls/future-innovator-sdg/]

Author response: Acknowledgements now read as:

‘Special thanks to all Stakeholders involved across Ireland and Uganda, as well as everyone who participated in sample and data collection including all patients, families, nurses, clinicians and laboratory scientists. The Health Innovation Hub Ireland (HIHI) is acknowledged and thanked for assisting with assessment of NeoSep-SAA at the Mercy University Hospital.’

Regarding Funding Statement in the online submission form:

Thank you, but there is no need to update the online Funding Statement as it accurately reflects the entirety of funding sources for the work. No funding was received from HIHI or Mercy University Hospital.

3. In the online submission form, you indicated that [The data underlying the results presented in the study are available from the Corresponding Authors.]. 

Author response: The ethical approval obtained for this work in both Uganda and Ireland did not specifically allow the sharing of anonymised individual patient data with individuals not associated with the project. Indeed, specific reference was made to keeping such data securely in locked offices and password-protected computers. Thus, we cannot comply with the requirement because public deposition would breach compliance with the protocol approved by our research ethics boards. An exemption is requested. We will of course make whatever collated data we can available to Requestors, but not if it breaches ethics permission.

Author response: Noted and done.

Comments to the Author:

1. Is the manuscript technically sound, and do the data support the conclusions?

Reviewer #1: Yes

Reviewer #2: Partly

2. Has the statistical analysis been performed appropriately and rigorously?

Reviewer #1: Yes

Reviewer #2: I Don't Know

3. Have the authors made all data underlying the findings in their manuscript fully available?

Reviewer #1: Yes

Reviewer #2: Yes

4. Is the manuscript presented in an intelligible fashion and written in standard English?

Reviewer #1: Yes

Reviewer #2: No

5. Review Comments to the Author

Author response: Issues 1-5 have been dealt with in response to individual Reviewer commentary.

Reviewer #1: Reviewer comments

Manuscript title: Diagnosis of neonatal and adult sepsis using a Serum Amyloid A lateral flow test.

Manuscript number: PONE-D-24-29224

Abstract

1. Page 2, line 35: Author states (AS)…’The difficulty in diagnosing neonatal sepsis……. sepsis-specific biomarkers.’

Reviewer comment (RC): The greatest difficulty is that the ‘Gold Standard’ for diagnosis of sepsis i.e. Blood culture is not always positive even in the presence of true sepsis and the long time it takes for the Blood culture result to be available even with BACTEC/BacTAlert.

A mention of this limitation must be made in the Introduction.

Author response: Noted with thanks. This has been clarified in the Introduction (page 3, R1: lines 76-79). 

‘Gold Standard’ test, blood culture, contribute to the difficulties in neonatal sepsis diagnosis. Blood cultures require large volumes of neonatal blood, specialised laboratory facilities and a long waiting period for results. Additionally, blood culture testing for sepsis has unacceptable low sensitivity and carries a high false negative risk [6]. 

2. Page 2, line 38: AS….’as a superior biomarker for the diagnosis of neonatal sepsis.’

RC: Acute phase reactants can only be used to indicate the probable diagnosis of sepsis or the absence of of sepsis (High NPV) in a neonate with commensurate clinical symptoms and signs. They are not a definitive test for diagnosis of sepsis. Throughout the manuscript, authors have mentioned the role of SAA as a diagnostic test. Even though I do not want to take away the enthusiasm of the research team and the authors, the manuscript must be modified where it is made abundantly clear that SAA is only a novel biomarker for the probable diagnosis of neonatal sepsis.

Author response: This is a very valid point. Serum Amyloid A might be thought of as a novel biomarker for diagnosis of probable neonatal sepsis because it is not currently routinely used in clinical diagnosis. However, the utility of SAA as a biomarker in neonatal sepsis diagnosis is not novel and has been suggested many times in literature (herein). Lack of suitable diagnostic systems for the detection of SAA has hindered its use as a biomarker in sepsis detection. Our work shows that SAA detection by appropriate LFT technology using microvolume blood collection aids in the probable diagnosis of neonatal sepsis. 

References: 

Chauhan N, Tiwari S, Jain U. Potential biomarkers for effective screening of neonatal sepsis infections: An overview. Microb Pathog. 2017;107:234-42. Epub 20170401. doi: 10.1016/j.micpath.2017.03.042. PubMed PMID: 28377234.

Balayan S, Chauhan N, Chandra R, Kuchhal NK, Jain U. Recent advances in developing biosensing based platforms for neonatal sepsis. Biosens Bioelectron. 2020;169:112552. Epub 20200825. doi: 10.1016/j.bios.2020.112552. PubMed PMID: 32931992.

Wu F, Hou XQ, Sun RR, Cui XJ. The predictive value of joint detection of serum amyloid protein A, PCT, and Hs-CRP in the diagnosis and efficacy of neonatal septicemia. Eur Rev Med Pharmacol Sci. 2019;23(13):5904-11. doi: 10.26355/eurrev_201907_18335. PubMed PMID: 31298341.

Bengnér J, Quttineh M, Gäddlin PO, Salomonsson K, Faresjö M. Serum amyloid A - A prime candidate for identification of neonatal sepsis. Clin Immunol. 2021;229:108787. Epub 20210625. doi: 10.1016/j.clim.2021.108787. PubMed PMID: 34175457.

Krishnaveni, P., Vanitha Gowda, M.N., Pradeep, G.C.M., 2016. Estimation of serum amyloid A protein in neonatal sepsis: a prospective study. Int. J. Medical Science and Public Health 2016; 5(8) 1665-1672.

Sharma V, Grover R, Priyadarshi M, Chaurasia S, Bhat NK, Basu S, et al. Point-of-Care Serum Amyloid A as a Diagnostic Marker for Neonatal Sepsis. Indian J Pediatr. 2023. 91(6):571-577. doi: 10.1007/s12098-023-04677-8.

3. Page 2, line 40: AS…. blood at patient-side….

RC: … at patient bedside….

Author response: Amended as suggested. Also amended throughout manuscript.

4. Page 2, line 40: AS….low resource environment…

RC: I suggest that ‘resource constrained’ or ‘resource limited’ be used throughout the manuscript rather than ‘low resource’.

Author response: Amended as suggested.

5. Page 2, line 46: AS…..(sensitivity: 37 %)…’.

RC: This sensitivity for CRP was as noted in the study cohort. However, abundant published literature has noted the sensitivity to be in the range as reported for the SAA.

Author response: The sensitivity we report for CRP is based on our study and statistical evaluations. We agree it is unexpectedly low, however this has been observed in publications before. 

References: 

Hedegaard SS, Wisborg K, Hvas AM. Diagnostic utility of biomarkers for neonatal sepsis--a systematic review. Infect Dis (Lond). 2015;47(3):117-24. Epub 20141218. doi: 10.3109/00365548.2014.971053. PubMed PMID: 25522182.

Cetinkaya M, Ozkan H, Köksal N, Celebi S, Hacimustafaoğlu M. Comparison of serum amyloid A concentrations with those of C-reactive protein and procalcitonin in diagnosis and follow-up of neonatal sepsis in premature infants. J Perinatol. 2009;29(3):225-31. Epub 20081211. doi: 10.1038/jp.2008.207. PubMed PMID: 19078972.

6. Page 2, line 51: AS….SDG 3.2 objectives.

RC: SDG can be expanded for the first time and subsequently can be mentioned as SDG throughout the text.

Author response: Amended as suggested.

Introduction

1. The introduction is rambling and requires to be made more focussed.

Author response: The Introduction has now been extensively revised and edited for focus. 

2. Why did the authors select SAA over Serum Procalcitonin and CRP?

Author response: SAA was selected due to the abundant evidence in the literature which supports its use a SAA as a biomarker for probable neonatal sepsis, and its potential superiority over other known biomarkers such as PCT or CRP- especially in Bengnér et al. (2021) (see references above). Consequently, we secured peer-reviewed Challenge funding to evaluate existing, rapid technology suitable for detection of SAA human blood and wanted to explore its potential as a ready-made solution for facilitating neonatal sepsis diagnosis in resource-limited environments.

3. Colorimetric card tests are available for CRP, PCT and are very efficiently being used in field out of hospital settings. Then why SAA?

Author response: There are a number of references to SAA being a superior biomarker for neonatal sepsis detection, especially its large dynamic range. In our study area, Uganda, CRP or PCT colorimetric card tests are not readily used, especially in non-clinical settings. We are not trying to displace any pre-existing biomarker of probable neonatal sepsis. We are attempting to demonstrate that robust LFT technology for SAA detection can be used for probable neonatal sepsis diagnosis in resource-limited environments. This has not previously been explored in detail.

4. In a suspected sepsis patient one would be collecting samples for Complete Blood Counts (CBC) and Blood culture. Thus minimally invasive sampling for SAA wold not be much of a benefit.

Author response: Absolutely correct and sampling for CBC and Blood culture should occur for suspected sepsis patients. However, in reality and in resource-limited or remote environments with no laboratory facilities such tests are not carried out or unavailable. Thus, a minimally-invasive SAA test would enable rapid testing of neonates with any suspicion of infection and facilitate decision-making to start antibiotic therapy or to refer patients to higher facilities. 

References: 

Yadav H, Shah D, Sayed S, Horton S, Schroeder LF. Availability of essential diagnostics in ten low-income and middle-income countries: results from national health facility surveys. Lancet Glob Health. 2021;9(11):e1553-e60. Epub 20211006. doi: 10.1016/s2214-109x(21)00442-3. PubMed PMID: 34626546; PubMed Central PMCID: PMCPMC8526361.

Rosa-Mangeret F, Dupuis M, Dewez JE, Muhe LM, Wagner N, Pfister RE. Challenges and opportunities in neonatal sepsis management: insights from a survey among clinicians in 25 Sub-Saharan African countries. BMJ Paediatr Open. 2024 Jun 17;8(1):e002398. doi: 10.1136/bmjpo-2023-002398. PMID: 38886111; PMCID: PMC11184178. 

(this is a new reference and not cited in manuscript)

5. In most neonatal units in neonates suspected to be having sepsis based on clinical features, antibiotic therapy would be initiated. Are the authors implying that use of SAA can limit starting antibiotics in symptomatic neonates before the results of blood/CSF/ urine culture are available. I think not. Thus the introduction must be about the ability of SAA to help a clinician stop antibiotics early.

Author response: Thank you for your comment, and we will clarify throughout the manuscript that detectio

---

## [Decision Letter · Decision Letter 1]

10 Nov 2024

PONE-D-24-29224R1Diagnosis of neonatal and adult sepsis using a Serum Amyloid A lateral flow test.PLOS ONE

Dear Dr. Doyle,

Thank you for submitting your manuscript to PLOS ONE. After careful consideration, we feel that it has merit but does not fully meet PLOS ONE’s publication criteria as it currently stands. Therefore, we invite you to submit a revised version of the manuscript that addresses the points raised during the review process.

We look forward to receiving your revised manuscript.

Kind regards,

Novel N. Chegou, Ph.D

Academic Editor

PLOS ONE

Journal Requirements:

Additional Editor Comments:

Please attend to the additional queries from Reviewer no 1

Reviewers' comments:

Reviewer's Responses to Questions

**Comments to the Author**

1. If the authors have adequately addressed your comments raised in a previous round of review and you feel that this manuscript is now acceptable for publication, you may indicate that here to bypass the “Comments to the Author” section, enter your conflict of interest statement in the “Confidential to Editor” section, and submit your "Accept" recommendation.

Reviewer #1: (No Response)

Reviewer #2: All comments have been addressed

2. Is the manuscript technically sound, and do the data support the conclusions?

Reviewer #1: Yes

Reviewer #2: Yes

3. Has the statistical analysis been performed appropriately and rigorously? 

Reviewer #1: Yes

Reviewer #2: I Don't Know

4. Have the authors made all data underlying the findings in their manuscript fully available?

Reviewer #1: Yes

Reviewer #2: Yes

5. Is the manuscript presented in an intelligible fashion and written in standard English?

Reviewer #1: Yes

Reviewer #2: Yes

6. Review Comments to the Author

Reviewer #1: Reviewer comments

Manuscript title: Diagnosis of neonatal and adult sepsis using a Serum Amyloid A lateral flow test.

Manuscript number: PONE-D-24-29224R1

Abstract

1. Page 2, line 38: Author States (AS)….’as a superior biomarker for the diagnosis of neonatal sepsis.’

Reviewer comment (RC): The authors have missed the point which I was trying to make.

The above sentence is required to be framed as …’as a superior biomarker for use in the sepsis screen for neonatal sepsis.’ OR ’….as a superior biomarker for the diagnosis of probable neonatal sepsis.’

Introduction

1. I thank the authors for their clarification of the queries raised.

The authors response to the undermentioned queries no. 2, 3, 4 from the previous review may be added in a concise form in the introduction, material & methods or discussion section.

2. Why did the authors select SAA over Serum Procalcitonin and CRP?

3. Colorimetric card tests are available for CRP, PCT and are very efficiently being used in field out of hospital settings. Then why SAA?

4. In a suspected sepsis patient one would be collecting samples for Complete Blood Counts (CBC) and Blood culture. Thus minimally invasive sampling for SAA wold not be much of a benefit.

Results

The explanation that ‘These 78 neonates were clinically diagnosed with sepsis by clinicians independent of the NeoSep-SAA result.’ Must be adequately be mentioned in the manuscript at a relevant place.

Reviewer #2: Reviewer has addressed all the comments and suggestions from the editor and the reviewers, and I have no additional comments.

7. PLOS authors have the option to publish the peer review history of their article (what does this mean?). If published, this will include your full peer review and any attached files.

Reviewer #1: **Yes: **Vishal Vishnu Tewari

Reviewer #2: **Yes: **Sana Mahtab

---

## [Author Response · Author response to Decision Letter 1]

14 Nov 2024

Manuscript title: Diagnosis of neonatal and adult sepsis using a Serum Amyloid A lateral flow test.

Manuscript number: PONE-D-24-29224R1

RESPONSE TO REVIEWERS (11 November 2024)

Reviewer #1: Reviewer comments

Manuscript title: Diagnosis of neonatal and adult sepsis using a Serum Amyloid A lateral flow test.

Manuscript number: PONE-D-24-29224R1

Abstract

1. Page 2, line 38: Author States (AS)….’as a superior biomarker for the diagnosis of neonatal sepsis.’

Reviewer comment (RC): The authors have missed the point which I was trying to make.

The above sentence is required to be framed as …’as a superior biomarker for use in the sepsis screen for neonatal sepsis.’ OR ’….as a superior biomarker for the diagnosis of probable neonatal sepsis.’

Author response 1: Done. Text now reads:

‘However, serum amyloid A (SAA) detection has potential as a superior biomarker for the diagnosis of probable neonatal sepsis.’

Introduction

1. I thank the authors for their clarification of the queries raised.

The authors response to the undermentioned queries no. 2, 3, 4 from the previous review may be added in a concise form in the introduction, material & methods or discussion section.

2. Why did the authors select SAA over Serum Procalcitonin and CRP?

3. Colorimetric card tests are available for CRP, PCT and are very efficiently being used in field out of hospital settings. Then why SAA?

Author response 2: Our pleasure. Reviewer Q2 and Q3 are addressed based on pre-existing text by addition of this sentence in R2 Introduction as follows:

‘Hence the choice of SAA over PCT and CRP as biomarker, with a location-compatible detection system, for our study.’

4. In a suspected sepsis patient one would be collecting samples for Complete Blood Counts (CBC) and Blood culture. Thus minimally invasive sampling for SAA wold not be much of a benefit.

Author response 3: Noted. We insert the text in yellow in R2 below, which highlights that it is not just lack of centralised testing but availability of ANY testing in resource-limited environments that is the problem, to address Reviewer Q4:

‘It has the potential to facilitate a rapid clinical decision in the absence of centralised test facilities or any existing patient-side biochemical analyses, and to initiate antibiotic therapy to combat sepsis.’

Results

The explanation that ‘These 78 neonates were clinically diagnosed with sepsis by clinicians independent of the NeoSep-SAA result.’ Must be adequately be mentioned in the manuscript at a relevant place.

Author response 4: Thank you. We have inserted this comment in yellow in Results as follows:

‘Additionally, 78 neonates within the CCG were clinically diagnosed with sepsis (Confirmed Sepsis: CS). Sepsis diagnosis was based on a suspicion of infection and combination of clinical symptoms or risk factors aided where possible by a CRP test, blood cultures and confirmed by an independent clinician’s diagnosis. Thus, these 78 neonates were clinically diagnosed with sepsis by clinicians independent of the NeoSep-SAA result. Of the 78 neonates, 72 (92%) neonates with a confirmed sepsis diagnosis tested positive in the NeoSep-SAA (Table 4).’

Reviewer #2: Reviewer has addressed all the comments and suggestions from the editor and the reviewers, and I have no additional comments.

Author response: Thank you.

---

## [Editor Report · Decision Letter 2]

15 Nov 2024

Diagnosis of neonatal and adult sepsis using a Serum Amyloid A lateral flow test.

PONE-D-24-29224R2

Dear Dr. Doyle

We’re pleased to inform you that your manuscript has been judged scientifically suitable for publication and will be formally accepted for publication once it meets all outstanding technical requirements.

Kind regards,

Novel Njweipi Chegou, Ph.D

Academic Editor

PLOS ONE
---

## [Editor Report · Acceptance letter]

21 Nov 2024

PONE-D-24-29224R2 

PLOS ONE

Dear Dr. Doyle, 

I'm pleased to inform you that your manuscript has been deemed suitable for publication in PLOS ONE. Congratulations! Your manuscript is now being handed over to our production team.

Kind regards, 

on behalf of

Prof Novel Njweipi Chegou 

Academic Editor

PLOS ONE